# Death of an Ex-Spouse: Lessons in Family Communication about Disenfranchised Grief

**DOI:** 10.3390/bs7020016

**Published:** 2017-03-24

**Authors:** Jillian A. Tullis

**Affiliations:** Department of Communication Studies, University of San Diego, 5998 Alcalá Park, Camino Hall 126, San Diego, CA 92110, USA; jtullis@sandiego.edu; Tel.: +1-619-260-6897

**Keywords:** autoethnography, loss, narrative, text message

## Abstract

The death of a loved one is an emotional-laden experience, and while grief and mourning rituals are less formal today in many communities, there remain some social norms for individuals to process loss. The death of an ex-family member, such as a former spouse, is more complicated and expectations for how to respond are fraught with uncertainty. While grief has been studied and is primarily understood as an individual cognitive process, scholars in sociology and communication are considering the ways in which grief and mourning are social and take place in dialogue with others. This manuscript explores Kenneth Doka’s concept of disenfranchised grief, which is “grief that is experienced when loss cannot be openly acknowledged, socially sanctioned, or publicly mourned” through the author’s experience of the death of her ex-husband. The narrative will recount how the author learned about her ex-husband’s death (via text message), and will challenge definitions of family and family communication about death and grief, particularly the communication strategies used to cope with this unique type of loss.

## 1. Introduction

“Did you hear that Tim P passed away yesterday?”

What the fuck?!?! I thought to myself. My ex-husband is dead.

So yeah, that was the moment I learned about the death of my former spouse. By text message. On a Sunday morning while I was lying in bed reading the news as I do most Sundays, with the dog nestled close to me. I had forgotten all about the pending message from my friend Pam, who was at my wedding 13 years ago. And there it was. Not even ten words situated in that familiar green text message bubble. Pam and I only talk on the phone a couple of times a year and until this Sunday in September the last time we spoke was in November around Thanksgiving. What the hell! I mean I always knew this day would come since Tim was more than two decades older than me. I had contemplated his dying, his death, and being a young widow. Once we were divorced, though, I had not given as much thought to what this moment would feel like. Any concerns I had about his death were only practical since we had one lingering financial tie. So I just stood in my bedroom, I was out of the bed by now, my mind racing. Looking back, I wish I had sat down, grounded myself a bit before making my next move. But I just stood there staring at my phone as if standing there would help this news make more sense. I replied to the text message:

“No. I didn’t. I hadn’t talked to him in years. Thanks for letting me know.”

But I was not thankful. I felt irritated. Why do people think it is okay to share this kind of news via text? I did what I normally do in these moments of shock and uncertainty. I called my mom.

## 2. Background and Methods

I study how people talk about dying and death and teach classes about how we communicate at the end of life. Not only am I infinitely fascinated by the subject, but I find it very life affirming. It gives me perspective and it teaches me so much about dying, but also about living and what makes a good life. Since this is my specialty, I have heard more than my share of stories about how people cope when their loved one dies. The narrative or story above, which I will return to in a moment, is one of those tales. It is a story about death and grief, but a particular type of grief, disenfranchised grief. It is also a story about families, how we define family and how we communicate in families about bereavement and the grief, or reactions that follow a loss. Grief is generally investigated and understood as an individual cognitive process, yet scholars in other fields of study investigate the ways in which grief and mourning are social and take place in conversation with others [1]. With this in mind, I use my own story to process this death, and to examine and understand communication with family about grief.

Before I return to the story about my reaction to the death of my ex-husband, I should explain a few things about how I present my story, also known in scholarly settings as autoethnography [2]. This approach to writing and research uses personal experience to understand aspects of culture. To accomplish this, the researcher or writer combines existing research about a topic with their personal story to offer new insights. Autoethnography, while useful and accessible to a broad audience, does not guarantee the same protections as other types of research. Therefore, names have been changed to protect the confidentiality of the people I write about. I have also fictionalized the story. This means that while the story really happened, I have changed other details in addition to the names, such as where people live, what kind of work they do, and their hobbies to further conceal their identities. While one of the major characters in this story is dead, I am still obligated to protect him and loved ones who are still alive and could read this article. Third, since I did not take notes during this experience, I have used my memory to recreate phone calls, text messages and face-to-face conversations. I have done my best to be as accurate as possible [3]. Now that I have explained my ethical obligations, some of the methodological issues, and how I have addressed them, let me return to the story.

## 3. Story Continues

“So, Tim P died”, I say stoically to my mom. I did not know how to say it so I just said it. “My ex-husband”, I added, because she did not respond immediately.

“Oh no!” I could hear her start to cry—a familiar response. It is almost the exact reaction she had when I told her one of my mentors had died suddenly. Instead of sympathizing with my mother’s feelings, I began to feel insecure and thought, ‘Oh shit! Maybe I’m supposed to be crying. Why am I not crying?’ Some tears welled up, as I stood in my bedroom talking with my mom, but they never fell. Crying just did not feel authentic; they would have been reactionary, a response to my mother’s emotions. She and my ex-husband were also closer in age than I was to Timothy, so I think it felt like a peer had died. I, on the other hand, could not feel much more than shock. Despite being in his early 60s, his death only seemed unexpected. And while I knew intellectually this was a tragedy, especially for his family, it did not feel that way emotionally for me. I figured whatever the circumstances, his death felt premature. Yet, I did not feel the sadness that comes with realizing someone you know and care about is permanently gone from your life because they have died.

“What happened?” she asked. “I shouldn’t be crying. He was your husband. I don’t even know why I’m crying.”

“It’s okay”, I said. I was actually glad one of us had some clear feelings about this news because I really did not. “I don’t know.” Truth be told, I had some guesses, but I was really afraid to say them aloud. An accident? I wondered. Cancer? Maybe he killed himself? No. No. No. He would not do that, I thought. Has to be something else. “Pam told me by text. I don’t know anything other than that he died yesterday.”

“Well that’s a crappy thing to do”, my mom said, referring to the text message. Her tone had changed and she was in full Mom-Mode, protective of her child’s emotional well-being and trying to take care of me by taking charge and giving me direction. “Call her. Find out what she knows. And call me back.”

“Okay.”

“Love you.”

“Love you too.” I end the call with my mom and search my contacts to dial Pam’s number. I feel a small sense of dread as the phone rings. She picks up by the second ring and sounds like she is expecting my call, which made me even more annoyed that she did not just ring me in the first place.

“Hey”, Pam said quietly with a knowing tone.

“Hey”, I said, trying to match her tone. “So, any idea what I happened?” I asked, hoping Pam could quell my worries that my ex-husband, the man I was partnered with for 12 years, did not die in a traumatic way. I realized, as Pam recounted what she knew, that this was my primary concern. The fact that my ex-husband was dead did not really seem like the most pressing issue, but how he died was my worry. This was the information I needed because it would help me know what his final days were like. I wanted to determine if he was suffering emotionally or physically, if he knew his death was coming, or if it was sudden. I think that because my work is driven by the desire to minimize people’s suffering at the end of life, I was more interested in having these details. Apparently, Pam was looped in on an email thread among some of our former co-workers, which helped me pull things together. It sounded like cancer.

My analytical, scholar brain kicked in as I contemplated the death of my ex-husband. Disenfranchised grief came immediately to mind. Kenneth Doka, a professor of gerontology, describes disenfranchised grief as “the grief that is experienced when a loss cannot be openly acknowledged, socially sanctioned, or publicly mourned” [4] (p. 160). He goes on to say that there are three ways this can occur. One is that the relationship between the deceased and the griever is not recognized. Consider, for example, the spouse who has dementia and does not always remember their partner and is not told about the death because they may not understand. In this case, the griever is perceived as incapable of experiencing the loss. The second is that the loss is not recognized as legitimate. Charles Corr (1999), a retired professor of philosophical studies, describes the death of a pet or a miscarriage that occurs early in a pregnancy as two types of loss that some might not view as real and worthy of strong emotions [5]. A third type of disenfranchised grief focuses on those relationships that are dismissed or not recognized. Divorced spouses are one example and this is why I wondered how I should communicate with others about this loss and how they would respond to me. The problem with disenfranchised grief, which is not present in those deaths that are socially and culturally recognized, is that problems can arise for the griever because there are few sources of support available to help facilitate mourning. Depression and prolonged grief are both concerns. Support groups for people whose ex-spouse has died are not abundant, in part because it is not socially acceptable or expected to grieve the death of someone they divorced. My experience fit into the third type of disenfranchised grief; following my ex-husband’s death, my relationship, our history together, was not recognized by my former in-laws and the possibility of disenfranchised grief only seemed more likely because I never felt like we reconciled once our marriage ended.

Now that I had some sense about what may have happened to Tim, that he had some illness that preceded his death, I started to wonder about the appropriate response to this news. Before I got off the phone with Pam, I asked her to pass along any information she might learn about a funeral, but not even Pam and her husband were sure they would attend. She was always more of my friend and had no contact with Tim after we split. I called my mom back and shared this new information and together we reflected on my life with Timothy.

By the time I got off the phone with my mom the second time, the feeling of shock had subsided some and I just started to feel sad for Tim. I think he was afraid to die because in all the years we were together he never seemed to accept his mortality. Despite the death of family members, Timothy never seemed interested in confronting the reality of his own death. In fact, the death of a niece may have contributed to his fear of dying. While we would occasionally talk about my research interests, Tim never seemed comfortable with the subject. Looking back, I suspect that embracing mortality would also require Tim to acknowledge the ways in which his behavior could contribute to an early death. If Tim were to accept that he would die, he might also feel compelled to reprioritize his goals and behavior. Since Tim seemed fearful, I was also really worried about him dying alone despite having a big family. I guess I was also worried he did not have a girlfriend in his life. If there were one term I would use to describe Tim it would be *serial monogamist*. As soon as I started to think about his family, my heart started to ache for his brother, who he was very close to, and his parents, who were both in their 90s. To bury your younger brother and your son was not right. I knew they would be heartbroken. I knew I would not attend any funeral or memorial service because there were too many unknowns, but moreover, my appearance after seven years apart would just detract from rather than comfort Tim’s family. I decided I would at least send a card to his parents expressing my condolences to them and my ex-brother and sisters in-law.

One week after first learning of Tim’s death, I had managed to gather a few more details from a woman Tim and I met at the NASCAR track we visited annually while we were married. Based upon what Deborah told me, I gathered that Tim had a growth on his neck and had surgery, but the cancer came back and by the time he was actively dying he was not able to talk and was eating through a tube. This was a special kind of punch to the gut.

In the last year of my doctoral program, I worked on a project with people who had terminal head or neck cancer. One morning I drove to the cancer center to interview a man whose cancer was advanced, his lips were so swollen from the disease that he was unable to speak for himself. His life partner, a woman he had been with for years, but was not married to, communicated on the man’s behalf. The efforts this couple made to participate in the study, driving early in the morning to meet with me before his appointment, moved me deeply. I encouraged everyone I saw to appreciate their ability to talk and smile. Later that night, after an emotionally taxing day, Tim locked me out of the house. This was really the beginning of the end of our marriage. I know this is probably going to sound strange, but in some weird way, Tim’s death seemed like the alternative ending to the movie of my life with him. While I could not have predicted how or when Timothy would die, I did expect to be with him in health and sickness until death.

After a week and no word from anyone in Tim’s family, I decided to do nothing. I did not even send a card to his parents. I talked with my brother about what to do, whether I should have any contact with Tim’s kids or one of his siblings and that is when it occurred to me that Tim’s family would support each other and it was the job of my family and friends to support me. While my mom, dad, and brother offered condolences and helped me process this strange life event, some of my friends were less available. I would call them, leave a generic message—I did not want to text or leave a voicemail announcing a death—and they just never called back. The majority of my other friends were thankfully very supportive, acknowledged and therefore validated that this was a legitimate loss that I might have significant, albeit conflicted emotions. Of the people I did speak with who knew Tim, none were surprised that he was dead. Sad for him and his family, but not really surprised. He lived a rough life. He smoked, he drank too much, and the drinking was a vicious cycle he could not break from, not even after we split. In fact, when I called the friend who was the best man at our wedding to tell him Tim had died, he confessed they had not kept in touch, in part because Tim was rarely sober. It is hard to stay friends or married to someone who is frequently intoxicated. Yet the loss of his marriage and friends did not appear to change his drinking behavior. So, while I felt some small comfort in knowing I was not the only one who could not make our made-family work, I still felt really ambivalent because we never reconciled. Tim never came around to seeing why our marriage ended. Even up until just weeks before our divorce was final, he was still talking about our family. The four of us: our two dogs and the two of us. Maybe it sounds naïve to say, but until he died, I had hoped he might call me and say he was sorry and that he finally realized his drinking was a problem.

Finding out someone you once cared about deeply has died via text message prompts a reaction of shock. Texting is a cold flat channel to receive such powerful information. I was worried about how I would feel as time passed. I wondered if I would eventually have a deep emotional breakdown or if I would feel ostracized and long for recognition if my former in-laws never contacted me. By the time an obituary was published, which my mom shared with me over the phone, attending any of the planned formal rituals that generally follow the death of an (current or former) intimate other was not feasible. This prompted additional questions about what exactly I should do to not only mourn the death of my ex-husband, but also get the closure I needed to further process the end of my marriage since the apology I hoped to receive was never going to come.

Tim’s death and the fact that no one from his family contacted me also raised questions about the meaning of family and how families communicate about dying and death. Friends and family members shape and legitimize romantic relationships and those in relationships look for affirmation from friends and family. I was surely a full-fledged member of Tim’s family at one time. They welcomed me and I cared about them, yet few family members had any contact with me when Tim and I split. In fact, when one family member called me during our separation and said, “You knew how he was when you married him”, I realized I could not count on his family to support me in any way during our separation or divorce. It was clear to me then, and again in his death, that despite more than a decade together, they would fully support Tim to my exclusion. The lack of contact in the years following our split seemed appropriate—they were *Team Tim P.*—but not reaching out to me after his death felt like an erasure—as if I never existed in the larger story of Tim’s life. His obituary offers another perspective and some insight into why this might have been the case. According to the obituary, published in his hometown paper, Tim had a girlfriend at the time of his death. It is also possible, that because Tim and I never reconciled, that his family simply honored his wishes to keep strict boundaries. I may never know. And in the absence of any contact with his family and any opportunity to participate in formal death rituals like a funeral, I would have to find my own path to grieving and mourning the death of my ex-husband.

It has been six months since I was notified, via text, of my ex-husband’s death. I talk freely about Tim when my life as a married person comes up and if people inquire about him today, I volunteer that he died. While I did visit his gravesite with my mom when I was home for Thanksgiving, I am still trying to come up with a ritual that would acknowledge and honor our relationship, as well as mark the end of Tim’s life. But this does not feel as urgent as it did six months ago. Perhaps this is the benefit of time. I did take one step towards catharsis. I rewrote Tim’s obituary. “Did the person who wrote this even know Tim!?” I lamented after my mom read the paltry sentences. The blurb was completely unsatisfying and it offered none of Timothy’s personality because it was just a listing of facts, none of which truly celebrated his life. So I rewrote it, in the form of a eulogy, partially to make up for the lack of flare, but in hopes of finding some conclusion. Here is an excerpt:
Timothy Patterson, known to many as Tim P, the man who declared at the age of 47 that he had consumed one million Budweisers, has died at the age of 61. He was most proud of having been a single father to his daughters and loved his granddaughters dearly. He also had a niece he was very close to, who was like a third daughter. Tim loved NASCAR, was mechanically inclined, and could fix anything. No one made better ribs than him and he had what I will call a very creative vocabulary. *Mofo* was one of his favorite words for those he loved and those he hated. To listen to Tim talk with his brother was like listening to two people speaking a foreign language. He was a reluctant dog owner when we were together, but only because he understood the pain of losing a pet. Nevertheless, he loved our pitbulls, Amber and Gus, so fully.

To be honest, in the days following the news that my ex-husband had died, I reflected on many positive memories, but as I looked over a few old photos I realized that the man I had some fond memories of had changed. I could see those changes in his face in those pictures. In older images of our first years together, Tim’s face conveyed a tranquil demeanor. In the later photos, Tim appeared less at peace and more distressed. Perhaps this look was the result of more daily drinking, unacknowledged depression following his retirement, or anxiety about our marriage. Maybe it was a confluence of these things, but it is clear now that I will likely never know. What is clear is that the first guy I wanted to know and the second not at all. So I knew that my reminiscing was not out of love. He had changed too much and I did not like the changes. Too many years had passed since my ex and I had any contact and the interactions we had in the years since our divorce were unpleasant, at least for me. In fact, they were crude reminders why we were no longer together. I knew I did not still love him. Perhaps this is the silver lining in the grey cloud of his death. I could truly move on and I did not need his family to affirm our life together any more than I needed them to help me mourn.

## 4. Discussion

According to the American Psychological Association, 40%–50% of married couples divorce [6]. While divorce is common, our rituals for grieving a divorced spouse are not clear. The role of families and how families communicate with us about our romantic relationships effect the development, maintenance, and dissolution of those relationships. They also have a deep influence on how we process loss through death. Families and friends can mean the difference between appropriate grief or a complicated, disenfranchised grieving experience. Despite the lack of contact with Tim’s family, I took steps to minimize my pain and process my feelings of loss through rewriting his obituary, talking with my family and friends (some who also knew Tim), and eventually visiting his grave. When we are able to support our friends and family whose grief runs the risk of being disenfranchised, we can best help them by offering them a listening ear. Talking and listening to the details of the relationship, its meaning, its challenges, and its triumphs are helpful to process a death. This is especially true if others are intentionally or unintentionally disenfranchising a griever through their messages or lack of communication. We can counter efforts to disenfranchise grief by listening when someone shares his or her story rather than shut down conversations that dismiss the relationship solely because it ended years earlier in divorce. Another suggestion is to offer to help that person create and participate in some ritual of closure. This could include visiting the deceased’s grave or creating some other event that formally signals an end. Given the variations on what constitutes “family” today, the possibility that grief could be disenfranchised is great and it is best to be prepared to offer support.

## 5. Conclusions

Death, whether sudden or expected, produces a range of emotions from shock to sadness to anger. How we learn about a death, although not well understood, seems to contribute to those reactions. In the last five years, I have received more bad news via social media and text message than ever before and I wonder what the implications are for my and others ability to grieve. While these communication technologies help us feel connected when we are alone, I cannot help but question if they offer the same quality of support as telephone or face-to-face interactions. The death of my ex-spouse illuminated these concerns, and prompted me to question the role of family in the grieving process, while challenging my thinking about how we define family. My goal in sharing my personal experience with the death of an ex-spouse and disenfranchised grief is to challenge our thinking about how we communicate about loss, but also encourage each of us to find effective ways support others when grief looms.

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
