# Peer review of "Death of an Ex-Spouse: Lessons in Family Communication about Disenfranchised Grief"

_behavsci, 2017, doi:10.3390/bs7020016_

Round 1

Reviewer 1 Report

To the author:

Thank you for allowing me to review this phenomenal manuscript. There are many strengths of this manuscript, including the quality of the writing, the use of autoethnography, the discussion of disenfranchised grief and what/who counts as family, and how others can support others with such grief. I also appreciate the observation about how the lack of contact from his family, post death, felt like erasure; the remarkable, re-written obituary; and then the recognitionor revelationthat you didnt need his family to affirm your life together or your mourning process.

I have only three MINOR suggestions for revision:

·      I would change the sentence, This means that while this is a 100% true story it really happenedI have changed to This means that while this story really happened, I have changed…”

·      I would change the sentence, “Memories are not full proof, I know, so I have done my best to be as accurate as possible, to I have done my best to be as accurate as possible.

·      The sentence, “home for Thanksgivings should be home for Thanksgiving.

This is an important and thoughtful essay, especially given, as you note, the prevalence of divorce and the lack of research on disenfranchised grief. I hope to see this manuscript published.

Author Response

Thank you for your generous review and comments. I have changed the manuscript in accordance with your recommendations.

Reviewer 2 Report

If you can have more than 3 keywords, I would suggest adding “grief” and “family”

“Appeared the text message in a green bubble” (pg. 1) sounds odd. Consider rephrasing

Line 26-Rephrase to “learned about the death of my ex-husband” to avoid repetition.

Line 28-The line about Pam seems either misplaced or needs more of a setup. Is she the one who sent the text? If not, I would suggest removing.

Remove “what is called” on lines 48-49

Rephrase “American culture about grief” on line 51-32

I suggest offering a brief definition of autoethnography after you first mention it on line 55

Line 47-Change “this is a 100% true story” to, “the event really happened” to streamline the writing

Line 67-“matter of stoically” sounds odd. Rephrase.

Line 68-Why did you add “My ex-husband” Wouldn’t your mom recognize his name?

Lines 76-77- “…his death only seemed unexpected and intellectually, but not 76 emotionally, tragic.” I suggest rephrasing this as it throws off the rhythm of your writing.

Lines 92-93-You change from past to present tense. Choose one or the other.

Line 126-Replace “his” with a clearer pronoun

Line 128-Remove “the”

Line 131-Add “phone” after “the”

Line 138-Why did Tim never accept his mortality? More information is needed.

Line 140-Change “monogamy” to “monogamist”

Line 141-Add a comma after “family” and “parents”

Line 157-Add a comma after “appointment”

Line 158-Add a comma after “day”

Line 158-Replace em dash with a period

Line 172-Add a comma after “knew”

Line 187-Place a period after “passed.” And rewrite the following two sentences as questions

Line 193-Add “was” after receive

Line 201-“anyway” should be separated into two words

Line 202-add a comma after “together”

Line 207-Rewrite “It is also possible that because Tim and I never reconciled that his family simply…” as  “It is also possible that, because Tim and I never reconciled, his family simply…”

Line 211-I would simply say “It’s been six months since I was notified, via text, of my ex-husband’s death.”

Line 213-Add a comma after “today”

Line 228-I would suggest replacing the em dash with a period and beginning a new sentence after.

Line 236-“there was a calm about Tim” sound odd. Rephrase.

Line 237-What made Tim seem more distressed? What were the signs? Show his decline.

Lines 242-243 sound odd, especially with the insertion of “or me.” Rephrase.

Author Response

Thank you for your close and careful read of my manuscript. I have responded to each of your suggestions below.

1. If you can have more than 3 keywords, I would suggest adding “grief” and “family”

 Thank you for this suggestion. I have not included the terms “grief” and “family” among the keywords because it is my understanding that searches on Google or library databases (Boolean searches) will include the title. I try to use synonyms when possible.

2. “Appeared the text message in a green bubble” (pg. 1) sounds odd. Consider rephrasing

The phrase was deleted and a reference was made on line 30 to address the comment #4 below as well. Hopefully this clarifies my relationship to the person sending the text messages and maintains the detail and imagery of receiving a text message.  

3. Line 26-Rephrase to “learned about the death of my ex-husband” to avoid repetition.

The phrase was edited to read: “The moment I learned about the death of my former spouse…”

4. Line 28-The line about Pam seems either misplaced or needs more of a setup. Is she the one who sent the text? If not, I would suggest removing.

Yes, you are correct. Pam is the person who sent the text message that opens the essay. I have attempted to make this more clear on lines 29-30.

5. Remove “what is called” on lines 48-49

Completed

6. Rephrase “American culture about grief” on line 51-32

I have rephrased this sentence to read, “I use my own story to not only process this death, but to examine and understand communication about grief. “ (Line 56) Although, I am not confident if I have addressed your concern since I am not clear from the suggestion if the sentence needs rephrasing because it is too broad, unclear, or some other issues. Hopefully this change addresses the concern.

7. I suggest offering a brief definition of autoethnography after you first mention it on line 55

Thank you for this very valuable suggestion. Beginning on line 64 I have added information about autoethnography.

8. Line 47-Change “this is a 100% true story” to, “the event really happened” to streamline the writing

Reviewer 1 also made this suggestion and it was changed (see line 65).

9. Line 67-“matter of stoically” sounds odd. Rephrase.

This was a typo, and has been changed on line 78 to read, “I say stoically”

10. Line 68-Why did you add “My ex-husband” Wouldn’t your mom recognize his name?

I added some additional details here to make it clear that I said this because she didn’t respond immediately (now line 79).

11. Lines 76-77- “…his death only seemed unexpected and intellectually, but not 76 emotionally, tragic.” I suggest rephrasing this as it throws off the rhythm of your writing.

Thank you for this suggestion. The sentence was cumbersome and was modified to read: “Despite being in his early 60s, his death only seemed unexpected. And while I knew intellectually this was a tragedy, especially for his family, it did not feel that way emotionally for me.” (now line 95-97)

12. Lines 92-93-You change from past to present tense. Choose one or the other.

I changed the tenses here (now line 112)

13. Line 126-Replace “his” with a clearer pronoun

Changed “his” to my “ex-husband’s” (now line 158)

14. Line 128-Remove “the”

15. Line 131-Add “phone” after “the”

Both completed: Line 159-160 and 163

16. Line 138-Why did Tim never accept his mortality? More information is needed.

I added some details about Tim’s state of mine and attitudes about dying at lines 172-178

17. Line 140-Change “monogamy” to “monogamist”

18. Line 141-Add a comma after “family” and “parents”

19. Line 157-Add a comma after “appointment”

20. Line 158-Add a comma after “day”

21. Line 158-Replace em dash with a period

Comments 17-21 above were all changed as suggested. Line numbers are as follows: This change was mad at line 180; Line 181 commas added; Line 197 comma added; Line 200 comma added; Line 200-201 both em dashes were removed.

22. Line 172-Add a comma after “knew”

The word “knew” appears on line 173 of the original document and the sentence reads as follows: “Of the people I did speak with who knew Tim none were surprised that he was dead.” In the revised document (line 214), the sentence now reads, “Of the people I did speak with who knew Tim, none were surprised that he was dead.” I hope this addresses your concern.

23. Line 187-Place a period after “passed.” And rewrite the following two sentences as questions

Thank you for this suggestion. I added a period after “passed” and to avoid posing questions I edited the next to sentences as follows: “I was worried about how I would feel as time passed. I wondered if I would eventually have a deep emotional breakdown or if I would I feel ostracized and long for recognition if my former in-laws never contacted me.”

24. Line 193-Add “was” after receive

25. Line 201-“anyway” should be separated into two words

26. Line 202-add a comma after “together”

27. Line 207-Rewrite “It is also possible that because Tim and I never reconciled that his family simply…” as  “It is also possible that, because Tim and I never reconciled, his family simply…”

28. Line 211-I would simply say “It’s been six months since I was notified, via text, of my ex-husband’s death.”

29. Line 213-Add a comma after “today”

30. Line 228-I would suggest replacing the em dash with a period and beginning a new sentence after.

Comments 24-30 all edited as recommended. Line references are as follows

Added at Line 238; Added at line 250; Added at line 251; Completed Line 256; Completed Line 260; Added Line 262; Deleted em dashes at Line 280 and 281

33. Line 236-“there was a calm about Tim” sound odd. Rephrase.

Line 289, the sentence now reads: In older photos of our first years together, Tim’s face conveyed a tranquil demeanor.

32. Line 237-What made Tim seem more distressed? What were the signs? Show his decline.

I added some additional details beginning at line 290. The section now reads: Perhaps this look was the result of more daily drinking, unacknowledged depression following his retirement, or anxiety about our marriage. Maybe it was a confluence of these things, but it is clear now that I will likely never know.

32. Lines 242-243 sound odd, especially with the insertion of “or me.” Rephrase.

Deleted “or me” at Line 306 

Thank you again for your helpful suggestions. I am confident the essay is better with your feedback.